# Cloud classification of ground-based infrared images combining manifold and texture features

Q. Luo[1], Y. Meng[1], L. Liu[1], X. Zhao[1], and Z. Zhou[1]

[1]College of Meteorology and Oceanology, National University of Defense Technology, Nanjing, 211101, China

5    *Correspondence to*: Z. Zhou (zhou_zeming@yahoo.com)

**Abstract.** Automatic cloud type recognition of ground-based infrared images is still a challenging task. A novel cloud classification method is proposed to group images into five cloud types based on manifold and texture features. Compared with statistical features in the Euclidean space, manifold features extracted on Symmetric Positive Definite (SPD) matrix space can describe the non-Euclidean geometric characteristics of the infrared image more effectively. The proposed method comprises three stages: pre-processing, feature extraction and classification. Cloud classification is performed by a Support Vector Machine (SVM). The datasets are comprised of the zenithal and whole-sky images taken by the Whole-Sky Infrared Cloud-Measuring System (WSIRCMS). Benefiting from the joint features, compared to the recent two models of cloud type recognition, the experimental results illustrate that the proposed method acquires a higher recognition rate with an increase of 2%-10% on the ground-based infrared datasets.

## 1. Introduction

The cloud has an essential impact on the absorption, scattering, emission of atmosphere, the vertical transport of heat, moisture and momentum (Hartmann et al., 1992; Chen et al., 2000). Cloud cover and cloud type can affect the daily weather and climate change through its radiation and hydrological effects (Isaac and Stuart, 1996; Liu et al., 2008; Naud et al., 2016). Therefore, accurate cloud detection and classification is necessary for meteorological observation. Nowadays, cloud cover changes and cloud type determination have been available through the ground-based sky imaging systems (Souzaecher et al., 2006; Shields et al., 2003; Illingworth et al., 2007). Different from traditional manual observation, ground-based sky-imaging devices can obtain continuous information of sky condition at a local scale with a high spatial resolution.

However, due to subject factors and a rough ground-based measuring system, the estimation of cloud cover and type may weaken their credibility (Tzoumanikas et al., 2012). Some attempts have been made to develop algorithms for cloud classification of ground-based images (Buch et al., 1995; Singh and Glennen, 2005; Cazorla et al., 2008; Heinle et al., 2010; Ghonima et al., 2012; Taravat et al., 2014; Zhuo et al., 2014). Wang and Sassen (2001) developed a cloud detection algorithm by combining ground-based active and passive remote sensing data to illustrate how extended-time remote sensing datasets can be converted to cloud properties of concern to climate research. Li et al. (2002) proposed a method for automatic classification of surface and cloud type using Moderate Resolution Imaging Spectro-radiometer (MODIS) radiance measurements, whose advantage lied in its independence of radiance or brightness temperature threshold criteria, and its

interpretation of each class was based on the radiative spectral characteristics of different classes. Singh and Glennen (2005) adopted the k-nearest neighbour (KNN) and neural network classifiers to identify cloud types with texture features, including autocorrelation, co-occurrence matrices, edge frequency, Law's features and primitive length. Calbó and Sabburg (2008) extracted statistical texture features based on the greyscale images, pattern features based on the spectral power function of images and other features based on the thresholded images for recognizing the cloud type with the supervised parallelepiped classifier. Heinle et al. (2010) chose 12 dimensional features mainly describing the colour and the texture of images for automatic cloud classification, based on the KNN classifier. Besides the statistical feature like the mean grey value of the infrared image, Liu et al. (2011) explored another six structure features to characterize the cloud structure for classification. Zhuo et al. (2014) validated that cloud classification may not perform well if the texture or structure features were employed alone. As a result, texture and structure features were captured from the colour image and then fed into a trained Support Vector Machine (SVM) (Cristlanini and Shawe-Taylor, 2000) to obtain the cloud type. Different from traditional feature extraction, Shi et al. (2017) proposed to adopt the deep convolutional activations-based features and provided a promising cloud type recognition result with a multi-label linear SVM model.

Automatic cloud classification has made certain achievements; however, the cloud classification of ground-based infrared images poses a great challenge to us. By far, few researches of cloud classification have been dedicated to the ground-based infrared images (Sun et al., 2009; Liu et al., 2011). Most recent methods conducted on the RGB visible images (Heinle et al., 2010; Zhuo et al., 2014; Li et al., 2016; Gan et al., 2017) cannot directly be exploited on the cloud type classification of infrared images due to the lack of colour information. Compared to colour images, infrared images can be obtained day and night continuously, which is important for practical application and analysis.

Nowadays, the Symmetric Positive Definite (SPD) matrix manifold has achieved success in many aspects, such as action recognition, material classification and image segmentation (Faraki et al., 2015; Jayasumana et al., 2015). As a representative of SPD matrix, the Covariance Descriptor (CovD) is a powerful tool to extract the feature of the image. It has several advantages. Firstly, it calculates the first-order and second-order statistics of the local patch. Secondly, it straightforward fuses various features. Thirdly, it is independent of the region size and has low dimensions. Fourthly, by subtracting the mean feature vector, the effect of the noisy samples is reduced to some degree. Finally, it is able to speed up the computation in images and videos using efficient methods (Tuzel et al., 2008; Sanin et al., 2013). Covariance matrices naturally form a connected Riemannian manifold. Although it proves effective, few investigations are pursued for the task of cloud classification with manifold features. The manifold feature vector can maintain these advantages of non-Euclidean geometric space and describe the image features comprehensively, so it is chosen for a try on the cloud classification. In this paper, a novel cloud classification method is proposed for ground-based infrared images. Manifold features, representing the non-Euclidean geometric structure of the image features, and texture features, expressing the image texture, are integrated for the feature extraction.

To exhibit the classification performance, we have compared the results with the other two models (Liu et al., 2015; Cheng and Yu, 2015), which are adapted for the classification task of infrared images. To make up for the weakness of the Local

Binary Patterns (LBP) that cannot describe the local contrast well, Liu et al. (2015) proposed a new descriptor called Weighted Local Binary Patterns (WLBP) for the feature extraction. And then the KNN classifier based on the chi-square distance was employed for cloud type recognition. Cheng and Yu (2015) incorporated statistical features and local texture features for block-based cloud classification. As Cheng and Yu (2015) reported, the method combining the statistical and uniform LBP features with the Bayesian classifier (Bensmail and Celeux, 1996) displayed the best performance in the 10-fold cross validation (Ripley, 2005) overall.

In this paper, the data and methodology of the method are described in Sect. 2. Section 3 focuses on the experimental results. Conclusions are summarized in Sect. 4.

## 2.Data and Methodology

In this section, the datasets and the methodology for cloud classification are introduced. The proposed method contains three main steps: pre-processing, feature extraction and classification. The framework is illustrated in Fig. 1.

### 2.1 Dataset and pre-processing

The datasets include the zenithal images and whole-sky images, which are gathered by the Whole-Sky Infrared Cloud Measuring System (WSIRCMS) (Liu et al., 2013). The WSIRCMS is a ground-based passive system that uses an uncooled microbolometer detector array of $320 \times 240$ pixels to measure downwelling atmospheric radiance in 8-14μm (Liu et al., 2011). A whole-sky image is obtained after combining the zenithal image and other images at eight different orientations. As a result, the zenithal image has a size of $320 \times 240$ pixels while the whole-sky image is of $650 \times 650$ pixels. The datasets are provided by National University of Defense Technology in Nanjing, China.

It is true that the clear sky background radiance in 8-14μm varies with time and zenith angle. The images of the datasets have been pre-processed in the consideration of this important factor. The clear sky radiance threshold in each image is calculated using the radiation transfer model (Liu et al., 2013). The real radiance $R$ at each pixel in each image is converted to the grey value $G_{pixel}$ between [0,255] with $G_{pixel} = R/(R_{temp} - R_{clear}) \times 255$, where $R_{clear}$ is the corresponding clear sky radiance threshold and $R_{temp}$ is the radiance corresponding to the real-time environment temperature. As a result, the effects of the clear sky background brightness temperature can be ignored, which means that this factor has little influence on the feature extraction of the images.

The cloud images used in the experiment are selected with the help of two professional meteorological observers with many years of observation experiences. The selection premise is that the chosen images should hold high visual quality and can be recognized by visual inspection. If an image is vague, it's hard for experts to justify its type. For the algorithm, it's difficult to extract effective features of a vague image, not to mention recognizing its cloud type. All infrared cloud images are labelled to construct the training set and testing set. To guarantee the golden-standard's confidence, only images labelled same by two meteorological observers are finally chosen as the dataset used in this study. Different from traditional cloud classification by observers, automatic cloud classification by the devices needs a new criterion for recognition. According to the morphology

and generating mechanism of the cloud, the sky condition is classified into five categories in this study (Sun et al., 2009): stratiform clouds, cumuliform clouds, waveform clouds, cirriform clouds and clear sky. The sky condition and its corresponding description are as shown in Table 1.

The zenithal dataset used in this study is selected from the historical dataset to assess the performance of the algorithm. To guarantee the reliability of true label of each image, the images without mixed cloud types are selected. The typical samples from each category are demonstrated in Fig. 2. As listed in Table 2, the zenithal dataset is comprised of 100 cloud images in each category.

The whole-sky dataset is obtained during July to October in 2014 at Changsha, China. Since the whole-sky image is obtained by combining the nine sub-images at different orientations, the division rules of the whole-sky dataset remain the same as that of the zenithal dataset. The whole-sky samples from each category are exhibited in Fig. 3. As listed in Table 2, the number of cases with stratiform clouds, cumuliform clouds, waveform clouds, cirriform clouds and clear sky is 246, 240, 239, 46 and 88, respectively.

As Fig. 4 shows, a pre-processing mask is provided on the whole-sky images, which is used to extract the region of interest (ROI) from the images, which is the areas of the clouds within the circle rather than the parts out of the circle. Different from the whole-sky images, all parts of the zenithal images are ROI. Thus, we implement the feature extraction directly on the original zenithal images.

## 2.2 Feature extraction

In addition to the manifold features proposed in this work, the texture features are also combined. The manifold features of the ground-based infrared image are extracted on the SPD matrix manifolds, and after that, they are mapped into the tangent space to form a feature vector in Euclidean space. The texture features represent the statistical information in Euclidean space; on the contrary, the manifold features describe the non-Euclidean geometric characteristics of the infrared image.

### 2.2.1 Texture features

In this paper, the Grey Level Co-occurrence Matrix (GLCM) is used to extract the texture features, including energy, entropy, contrast and homogeneity (Haralick et al., 1973). Each matrix element in the GLCM represents the joint probability occurrence $p(i,j)$ of pixel pairs with a defined direction θ and a pixel distance d having grey level values $i$ and $j$ in the image.

$$\text{GLCM} = \begin{bmatrix} p(0,0) & p(0,1) & p(0,2) & \dots & p(0,k-1) \\ p(1,0) & p(1,1) & p(1,2) & \dots & p(1,k-1) \\ \vdots & \vdots & \vdots & \dots & \vdots \\ p(k-1,0) & p(k-1,1) & p(k-1,2) & \dots & p(k-1,k-1) \end{bmatrix}_{k \times k}. \quad (1)$$

The energy measures the uniformity and texture roughness of the grey level distribution:

$$\text{Energy} = \sum_{i=0}^{k-1} \sum_{j=0}^{k-1} p(i,j)^2. \quad (2)$$

The entropy is a measure of randomness of grey level distribution:

$$\text{Entropy} = -\sum_{i=0}^{k-1}\sum_{j=0}^{k-1} p(i,j)\log_2 p(i,j). \tag{3}$$

The contrast is a measure of local variation of grey level distribution:

$$\text{Contrast} = \sum_{i=0}^{k-1}\sum_{j=0}^{k-1}(i-j)^2 p(i,j)^2. \tag{4}$$

The homogeneity measures the closeness of the distribution of elements in the GLCM to the GLCM diagonal:

$$\text{Homogeneity} = \sum_{i=0}^{k-1}\sum_{j=0}^{k-1}\frac{p(i,j)}{1+|i-j|}. \tag{5}$$

As the number of intensity levels $k$ increases, the computation of the GLCM increases strongly. In this work, $k$ is set with 16 and then the texture features are obtained by calculating four GLCMs with d $= 1$ and $\theta = 0°$, $45°$, $90°$, $135°$, respectively. To alleviate the complexity, reduce the dimension and keep rotation invariance, four mean features of four GLCMs with $\theta = 0°$, $45°$, $90°$, $135°$ are obtained as the final texture features. In the experiments, we find that these texture features are significant for the cloud classification of the ground-based infrared image.

### 2.2.2 Manifold features

The manifold features are attained by two steps: computing the regional CovD and mapping the CovD into its tangent space to form a feature vector.

**Step 1:** Computing the regional CovD

Suppose the image $I$ is of the size $W \times H$, its d-dimensional features containing greyscale and gradient at each pixel are computed, which compose the feature image $F$, whose size is $W \times H \times d$:

$$F(x,y) = f(I,x,y), \tag{6}$$

where the feature mapping $f$ is defined as:

$$f = \left[ I(x,y) \quad |I_x| \quad |I_y| \quad \sqrt{|I_x|^2 + |I_y|^2} \quad |I_{xx}| \quad |I_{yy}| \right]^{\mathrm{T}}, \tag{7}$$

In which $(x,y)$ denotes the location, $I(x,y)$ denotes the greyscale. $|I_x|$, $|I_y|$, $|I_{xx}|$ and $|I_{yy}|$ represent the first and second order derivative in the direction of $x$ and $y$ at each pixel, respectively. $\sqrt{|I_x|^2 + |I_y|^2}$ denotes the modulus of gradient.

For the feature image $F$, supposing it contains $n = W \times H$ points of d-dimensional features $\{f_k, k = 1,2,\dots,n\}$. Its CovD is a $d \times d$ covariance matrix, computed by Eq. (8):

$$C = \frac{1}{n-1}\sum_{k=1}^{n}(f_k - \mu)(f_k - \mu)^{\mathrm{T}}, \tag{8}$$

where $\mu = \frac{1}{n}\sum_{k=1}^{n} f_k$, which represents the feature mean vector.

The CovD can fuse multiple dimensional features of the image and express the correlations between different features. Besides, since the CovD is symmetric, it is only $d(d+1)/2$ dimensional. If we convert the CovD into a feature vector to describe the image, its dimension is $n \times d$, which needs a high computation cost for cloud classification.

**Step 2:** Obtaining the feature vector by mapping the CovD into its tangent space

Generally speaking, the manifold is a topological space that is locally equivalent to a Euclidean space. The differential manifold has a globally defined differential structure. Its tangent space $T_X M$ is a space formed by all possible tangent vectors at a given point $X$ on the differential manifold. For the Riemannian manifold $M$, an inner product is defined in its tangent space. The shortest curve between two points on the manifold is called the geodesic and the length of the geodesic is the distance between two points.

All SPD matrices form a Riemannian manifold. Suppose $S^d$ is a set of all $n \times n$ real symmetric matrices: $S^d = \{A \in M(d): A^T = A\}$, where $M(d)$ represents the set of all $d \times d$ matrices, so that $S_{++}^d = \{A \in S^d: A > 0\}$ is the set of all $d \times d$ SPD matrices, which construct a $d(d + 1)/2$ dimensional SPD manifold. According to the operation rules of the matrix, the set of the real symmetric matrix is a vector space while the real SPD matrix space is a non-Euclidean space. A Riemannian metric should be given to describe the geometric structure of the SPD matrix and to measure the distance of two points on $S_{++}^d$.

Geodesics on the manifold are related to the tangent vectors in the tangent space. Two operators, exponential map $\exp_X(\cdot): T_X M \to M$ and the logarithm map $\log_X(\cdot) = \exp_X^{-1}(\cdot): M \to T_X M$, are defined over differentiable manifolds to switch between the manifold and its tangent space at $X$. As illustrated in Fig. 5, the tangent vector $v$ is mapped to the point $Y$ on the manifold through the exponential map. The length of $v$ is equivalent to the geodesic distance between $X$ and $Y$ due to the property of the exponential map. Conversely, a point on the manifold is mapped to the tangent space $T_X M$ through the logarithm map. As point $X$ moves along the manifold, the exponential and logarithm maps change. The details can be referred in Harandi et al. (2012).

For $S_{++}^d$, the logarithm and exponential maps are given by:

$$\log_X(Y) = X^{\frac{1}{2}}\log(X^{-\frac{1}{2}}YX^{-\frac{1}{2}})X^{\frac{1}{2}}, \tag{9}$$

$$\exp_X(y) = X^{\frac{1}{2}}\exp(X^{-\frac{1}{2}}yX^{-\frac{1}{2}})X^{\frac{1}{2}}, \tag{10}$$

where $\log(\cdot)$ and $\exp(\cdot)$ are the matrix logarithm and exponential operators, respectively. For SPD matrices, they can be computed through Singular Value Decomposition (SVD). If we let $\mathrm{diag}(\lambda_1, \lambda_2, \dots, \lambda_d)$ be a diagonal matrix formed from real values $\lambda_1, \lambda_2, \dots, \lambda_d$ on diagonal elements and $A = U\mathrm{diag}(\lambda_i)U^T$ be the SVD of the symmetric matrix $A$, then

$$\log(A) = \sum_{r=1}^{\infty}\frac{(-1)^{r-1}}{r}(A - I)^r = U\mathrm{diag}(\ln(\lambda_i))U^T, \tag{11}$$

$$\exp(A) = \sum_{r=0}^{\infty}\frac{1}{r!}A^r = U\mathrm{diag}(\exp(\lambda_i))U^T, \tag{12}$$

where $I$ is an identity matrix on manifolds.

The manifold can be embedded into its tangent space at identity matrix $I$. Thus, based on the bi-invariant Riemannian metric (Arsigny et al., 2008), the distance between two SPD matrices $A$, $B$ is $d(A, B) = \|\log(A) - \log(B)\|_2$, where $\log(\cdot)$ denotes matrix logarithm operator. Since symmetric matrices (equivalently tangent spaces) form a vector space, the classification tools in the Euclidean space (SVM, KNN and so on) can be seamlessly employed to deal with the recognition problem.

The logarithmic operator is valid only if the eigenvalues of the symmetric matrix are positive. When no cloud is observed in the clear sky, the CovD of the image features could be non-negative definite, and in this case, it needs to be converted to a SPD matrix. We can formulate it as an optimization problem (Harandi et al., 2015):

$$A^* = \arg\min_{A} \|C - A\|_F, \text{s.t.} A + A^T > 0, \tag{13}$$

where $C$ is a CovD and $A^*$ is the closest SPD matrix to $C$.

For a SPD matrix $A$, its log-Euclidean vector representation $a \in \mathbb{R}^m$, $m = d(d+1)/2$, is unique and can be represented as $a = Vec(\log(A))$. Let $B = \log(A)$, $B \in S^d$ and

$$B = \begin{bmatrix} b_{1,1} & b_{1,2} & b_{1,3} & \dots & b_{1,d} \\ b_{2,1} & b_{2,2} & b_{2,3} & \dots & b_{2,d} \\ \vdots & \vdots & \vdots & \dots & \vdots \\ b_{d,1} & b_{d,2} & b_{d,3} & \dots & b_{d,d} \end{bmatrix}_{d \times d}, \tag{14}$$

which lies in the Euclidean space. Since $B$ is symmetric, we can rearrange it into a vector by vectorizing its upper triangular matrix:

$$a = Vec(B) = \left[ b_{1,1}, \sqrt{2}b_{1,2}, \cdots, \sqrt{2}b_{1,d}, b_{2,2}, \sqrt{2}b_{2,3}, \cdots, b_{d,d} \right]^T. \tag{15}$$

Vector $a$ is defined as the manifold features. Since $f$ is a 6-dimensional feature mapping in the experiment, the manifold feature vector $a$ to represent the cloud image is $6 \times (6+1)/2 = 21$ dimensions. The mapped feature vector can reflect the characteristics of its corresponding SPD matrix on matrix manifolds. Thus, manifold features can describe the non-Euclidean property of the infrared image features to some degree.

### 2.2.3 Combining manifold and texture features

As described in Sect. 2.2.1 and 2.2.2, manifold and texture features can be extracted and integrated to represent the ground-based infrared images. For an image, its four features including energy, entropy, contrast and homogeneity from GLCM, express its texture, while 21-dimensional manifold features describe the non-Euclidean geometric characteristics. The manifold and texture features are combined to form a feature vector to represent the image. Thus, the joint features of the infrared image have a total of 25 dimensions.

### 2.3 Classification

### 2.3.1 Support vector machine

The classifier used in this paper is the SVM (Cristlanini and Shawe-Taylor, 2000), which exhibits prominent classification performance in the cloud type recognition experiments (Zhuo et al., 2014; Li et al., 2016; Shi et al., 2017). In machine learning, SVMs are supervised learning models. An SVM model is a representation of the examples as points in the Reproducing Kernel Hilbert Space, mapped so that the examples of the separate categories are divided by a clear gap that is as wide as possible. New examples are then mapped into that same space and predicted to belong to a category based on which side of the gap they

fall. As Fig. 6 shows, given a set of two-class training examples (denoted by × and o, respectively), the key problem is to find the optimal hyperplane to do the separation: $w^T x + b = 0$, where $w$ is a weight vector and $b$ is a bias, and an SVM training model with the largest margin $2/\sqrt{w^T w}$ is built. The support vectors are the samples on the dotted lines. The optimization classification hyperplane is determined by the solid line. The test examples are assigned to one category or the other based on this model, making it a non-probabilistic binary linear classifier. In this work, we apply a simple linear function as the mapping kernel, which is validated by the cloud classification experiments.

### 2.3.2 Multi-class support vector machine method

For a multi-class task, one binary SVM classifier is constructed for every pair of distinct classes, and so, all together $c(c - 1)/2$ binary SVM classifiers are constructed. For an unknown-type sample, it will be input into these binary classifiers and each classifier makes its vote, thus $c(c - 1)/2$ independent output labels are obtained. The most frequent label is the sample's type. $c$ is 5 in this paper and the final result is determined by the voting policy.

### 3. Experiments and discussions

In this section, we validate which features are chosen and report experimental results to assess the performance of the proposed cloud classification method. Different from a deterministic case, the training samples of the experiments are chosen randomly. The effects of the proposed features are first tested by conducting the 10-fold cross validation (Li et al., 2016; Gan et al., 2017) 50 times on two datasets, respectively, and average values are taken as final results. In the 10-fold cross validation, each dataset is divided into 10 subsets with the same size at random. One single subset is used for validation in turn and the other 9 parts are taken as the training set. The results of 10-fold cross validation with different features are given in Table 3. As Table 3 illustrates, the overall accuracy of texture, manifold and combined features achieves 83.49%, 96.46% and 96.50% on the zenithal dataset while 78.01%, 82.38% and 85.12% on the whole-sky dataset, respectively. It can be seen that the texture or manifold features alone don't achieve a better performance than the joint features, which not only inherit the advantage of the texture features, but also own the characteristic of manifold features. On the whole, the method using the joint features performs best in the cross validation.

Naturally, combined features are used for the cloud type recognition. In the experiment, each dataset is grouped into the training set and testing set. The training set is selected randomly from each category in accordance with a certain proportion 1/10, 1/2 and 9/10, respectively and the rest part forms the testing set. Each experiment is repeated for 50 times to reduce the accidental bias and the average accuracy is regarded as the final results of classification to evaluate the performance of the method.

To exhibit the recognition performance of the proposed method, we also compare with the other two models (Liu et al., 2015; Cheng and Yu, 2015) to assess its performance in this experiment. Liu's model employs WLBP feature with the KNN classifier based on the chi-square distance while Cheng's method adopts the statistical and uniform LBP features with the Bayesian classifier. Note that we extract the statistical features from the greyscale images rather than from the RGB images so

that the statistical features have only 8 dimension, as a result, without extra colour information provided, both of the two methods are adaptable to the infrared images.

### 3.1 Results of the zenithal dataset

The first experiment is performed on the zenithal dataset. Table 4 reports the overall recognition rates of the proposed method and the other methods. The proposed method attains the best results, with at least 2.5% improvement over Liu's method and over 9.5% higher than Cheng's method. Meanwhile, the proposed method demonstrates a more stable and more superior performance than the other two methods, even when 1/10 of the dataset is treated as the training set. In this case, the proposed method is up to 90.85% on the overall accuracy while the other two methods achieve 81.30% and 81.64%, respectively. That means discriminative features used for classification can be gained even with what would normally be regarded as limited training data. Although only three cases are given when the fractions of training set are 1/10, 1/2 and 9/10, they can represent most cases. In general, with the increase in the number of training samples, the overall accuracy of testing samples will increase until it holds stable, which is in line with the results in Table 4. As a result, as more representative images are used for training, there is no doubt that the recognition rate will be improved.

In Fig. 7, the classification results of the proposed method are demonstrated in the form of the confusion matrix (Zhuo et al., 2014; Liu et al., 2015; Li et al., 2016) when 1/2 of the dataset constructs the training set while the rest 1/2 is used for testing. In the confusion matrix, each row of the matrix represents an actual class while each column represents the predicted class given by SVM. For example, the element in the second row and third column is the percentage of cumuliform clouds misclassified as waveform clouds. Therefore, the recognition rate for each class is in the diagonal of the matrix. The discrimination rate of stratiform clouds is up to 100%, which indicates that stratiform clouds have the most significant features to be distinguished among five cloud types. Likewise, the results of the other four cloud types achieve over 93%. It is shown that a rather high accuracy of each cloud type is reached, which means the proposed method performs well in classifying the ground-based infrared zenithal images on the whole when compared to the analysis of meteorological experts.

### 3.2 Results of the whole-sky dataset

The second experiment is performed on the whole-sky dataset, which is more challenging because there exists larger inner-class difference than that of the zenithal dataset. The experimental configuration retains the same in Sec. 3.1. Table 5 lists the results of different methods. It is illustrated that the proposed method gains the overall accuracy of 78.27%, 83.54% and 85.01% as the proportion of the training set varies. In comparison, Liu's method achieves 73.58%, 80.55% and 81.31% while Cheng's method achieves 66.99%, 67.36% and 68.18%, correspondingly. Comparing to the other two methods, the experimental results indicate the effectiveness of the proposed method with an obvious improvement in the accuracy. Similarly, the cases where the fractions of data reserved for training constitute1/10, 1/2 and 9/10 of the total, can represent most cases of classification and are chosen to represent a wide range of possible training scenarios. Generally, the rise in the number of training samples

makes the overall accuracy improve, which is in line with the results in Table 5. In a nutshell, a training set with more representative images can further promote the classification accuracy.

Figure 8 displays the confusion matrix of the whole-sky dataset when 1/2 for training is used. The number of each category in the training set is 123, 120, 120, 23 and 44, respectively and the remaining part is treated as the testing set. It is demonstrated that stratiform clouds and clear sky possess obvious characteristics for classification while cumuliform, waveform and cirriform clouds pose a great challenge for a high accuracy of classification. Cirriform clouds are likely to be confused with the clear sky and about 15.22% of cirriform cloud images are misclassified as the clear sky in the experiment. In the whole-sky image, when it is on the condition of cirriform clouds, the area of cirriform clouds may be just a fraction of the whole sky, making it hard to be distinguished correctly. What's more, multiple cloud types could exist in the whole-sky condition, which may result in a relatively low accuracy of the single-type classification, like cumuliform, waveform and cirriform clouds.

There are some misclassifications, just as demonstrated in Fig. 9. Figure 9(a) shows that stratiform clouds are recognized as waveform clouds. It can be seen that the cloud base has a little fluctuation and makes it similar to the waveform cloud. Figure 9(b) shows that cumuliform clouds are recognized as waveform clouds. We can distinguish it as waveform clouds by the shape but the strong vertical motion of cumuliform clouds makes it hard to differ from waveform clouds. Figure 9(c) shows that cumuliform clouds are recognized as cirriform clouds. In this image, besides cumuliform clouds, a little cirriform clouds can also be found. Figure 9(d) shows that waveform clouds are recognized as cumuliform clouds. It can be seen that both waveform and cumuliform clouds coexist in the sky. Figure 9(e) shows that cirriform clouds are recognized as cumuliform clouds. It is admitted that the whole-sky dataset is more complicated than the zenithal dataset as the weather conditions change.

## 4. Conclusions

In this paper, a novel cloud classification method of the ground-based infrared images, including the zenithal and whole-sky datasets, is proposed. Besides the texture features computed from the GLCM, manifold features obtained from the SPD matrix manifold are combined together. With the joint features, the proposed method can improve the recognition rate of the cloud types. On one hand, the joint features can inherit the advantages of the statistical features, which represent texture information in Euclidean space; on the other hand, the manifold features on the matrix manifold can describe the non-Euclidean geometric structure of the image features and thus the proposed method can benefit from it for a high classification precision. The CovD is calculated by extracting 6-dimensional features including greyscale, first-order and second-order gradient information, and the mean values are subtracted from the feature vectors, which may improve the recognition performance to some extent, since it can remove the noises of the infrared images. The manifold feature vector is produced by mapping the SPD matrix into its tangent space and afterwards the combined feature vector is adopted for cloud type recognition with SVM. With different fractions that the training set occupies, it is validated that in most cases the proposed method outperforms the other two methods (Liu et al., 2015; Cheng and Yu, 2015). As a whole, the improvement of the proposed method is between 2% and 10%. To some degree, it may not be a great improvement, but we have validated that the introduction of manifold features is effective and can achieve some success, it is worthy doing more work in this field to promote its development.

In future work, more suitable image features like Gabor or wavelet coefficients (Liu and Wechsler, 2002) can be incorporated into the SPD matrix and the classification would be performed directly on the manifolds to improve the recognition rate further. Besides, feature extraction using deep learning method such as convolutional neural networks can be taken into account to increase the classification accuracy. What's more, the addition of the brightness temperature, or the height information obtained from the laser ceilometer might be helpful for the improvement of the cloud type recognition accuracy. It is found that the proposed method is effective to satisfy the requirement of the cloud classification task on both zenithal and whole-sky datasets. The complex sky condition with multiple cloud types should arise our concern in the next work.

## 5. Code availability

The code of the proposed method can be available via email to qixiang_luo@aliyun.com.

## 6. Data availability

The two ground-based infrared cloud datasets used in this paper can be available via email to liuleidll@gmail.com.

*Acknowledgements.* This work is in part supported financially by the National Natural Science Foundation of China under Grant No.61473310, No.41174164, No.41775027 and No. 41575024.

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

**Table 1. The sky condition classes and corresponding description.**

| Sky condition classes | Description | Cloud types |
|---|---|---|
| Stratiform clouds | Horizontal, layered clouds that stretch out across the sky like a blanket | St、As、Cs (Sc、Ac、Cb、Ns) |
| Cumuliform clouds | Thick clouds that are puffy in appearance, like large cotton balls | Cu、Cb |
| Waveform clouds | Thin or thick clouds occurring in sheets or patches with wavy, rounded masses or rolls | Sc、Ac、Cc |
| Cirriform clouds | Thin clouds; very wispy and feathery looking | Ci |
| Clear sky | Clear | No clouds |

**Table 2. The numbers of each class on two datasets.**

| Sky condition classes | Zenithal | Whole-sky |
|---|---|---|
| Stratiform clouds | 100 | 246 |
| Cumuliform clouds | 100 | 240 |
| Waveform clouds | 100 | 239 |
| Cirriform clouds | 100 | 46 |
| Clear sky | 100 | 88 |
| Total | 500 | 859 |

**Table 3. The 10-fold cross validated classification accuracy (%) on two datasets.**

|  | Zenithal | Whole-sky |
|---|---|---|
| Texture features | 83.49 | 78.01 |
| Manifold features | 96.46 | 82.38 |
| Combined features | **96.50** | **85.12** |

**Table 4. The overall classification accuracy (%) on the zenithal dataset. 1/10, 1/2 and 9/10 are the certain proportions of the training set selected randomly from each category, and the rest forms the testing set correspondingly.**

|  | 1/10 | 1/2 | 9/10 |
|---|---|---|---|
| Liu's method | 81.64 | 92.24 | 93.48 |
| Cheng's method | 81.30 | 81.92 | 81.32 |
| Proposed method | **90.85** | **95.98** | **96.36** |

**Table 5. The overall classification accuracy (%) on the whole-sky dataset. 1/10, 1/2 and 9/10 are the certain proportions of the training set selected randomly from each category, and the rest forms the testing set correspondingly.**

|                 | 1/10      | 1/2       | 9/10      |
|-----------------|-----------|-----------|-----------|
| Liu's method    | 73.58     | 80.55     | 81.31     |
| Cheng's method  | 66.99     | 67.36     | 68.18     |
| Proposed method | **78.27** | **83.54** | **85.01** |

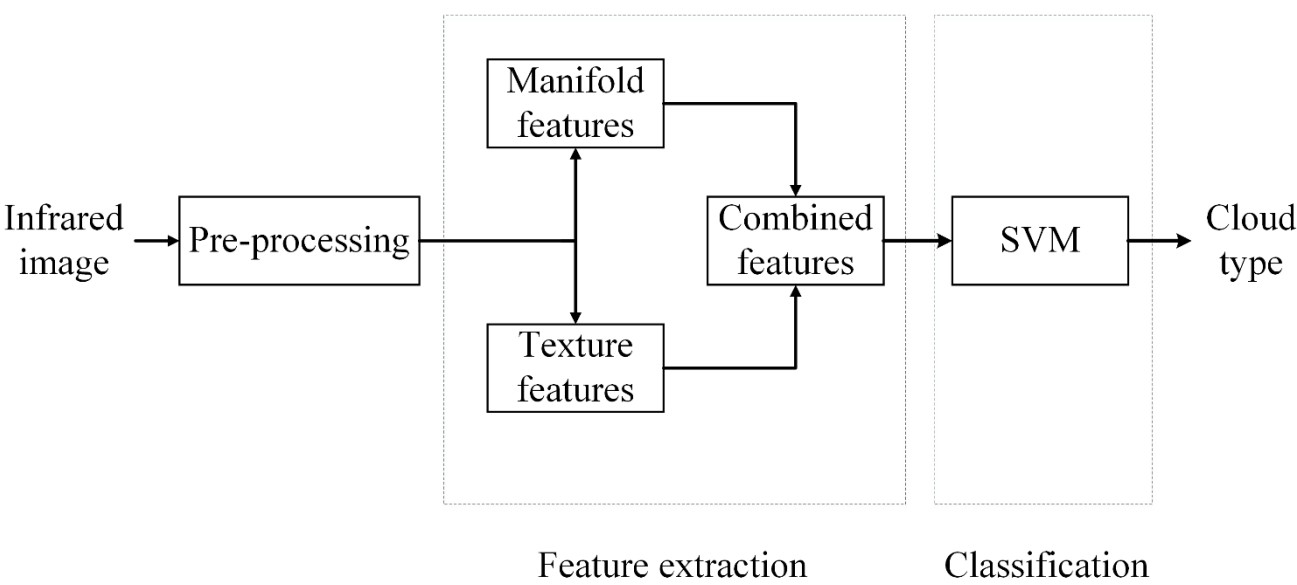

**Figure 1: System framework.**

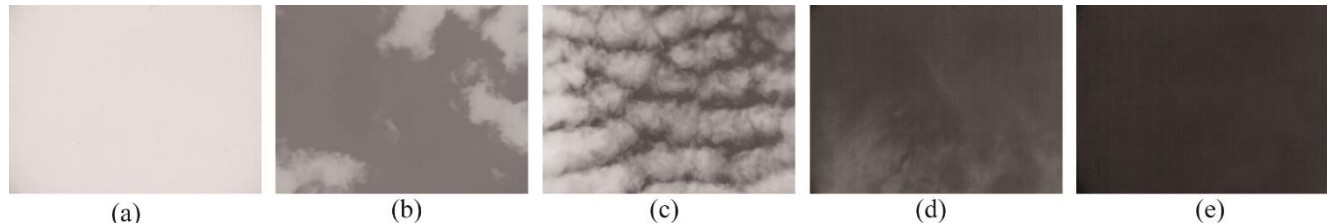

**Figure 2: Cloud samples from the zenithal dataset. (a) stratiform clouds, (b) cumuliform clouds, (c) waveform clouds, (d) cirriform clouds and (e) clear sky.**

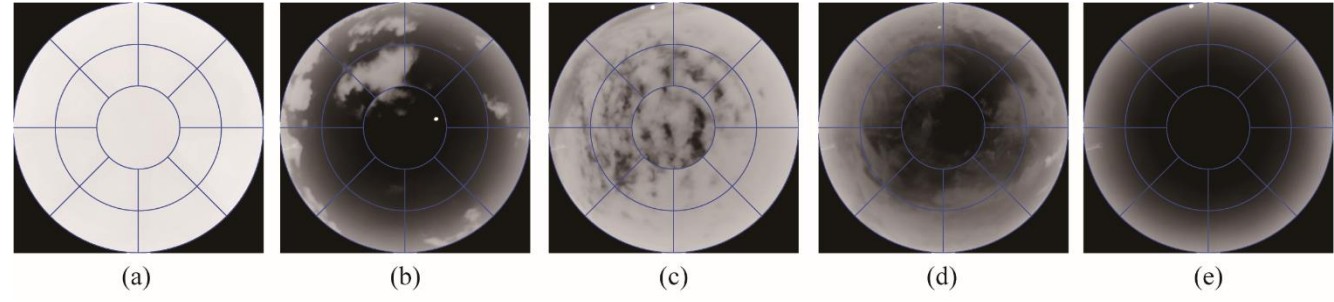

**Figure 3: Cloud samples from the whole-sky dataset. (a) stratiform clouds, (b) cumuliform clouds, (c) waveform clouds, (d) cirriform clouds and (e) clear sky.**

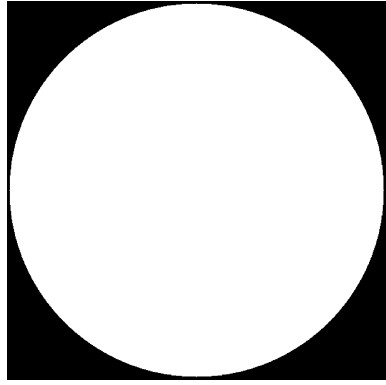

**Figure 4: The mask of the whole-sky images. The area within the circle is the ROI, and the area outside the circle is not the ROI.**

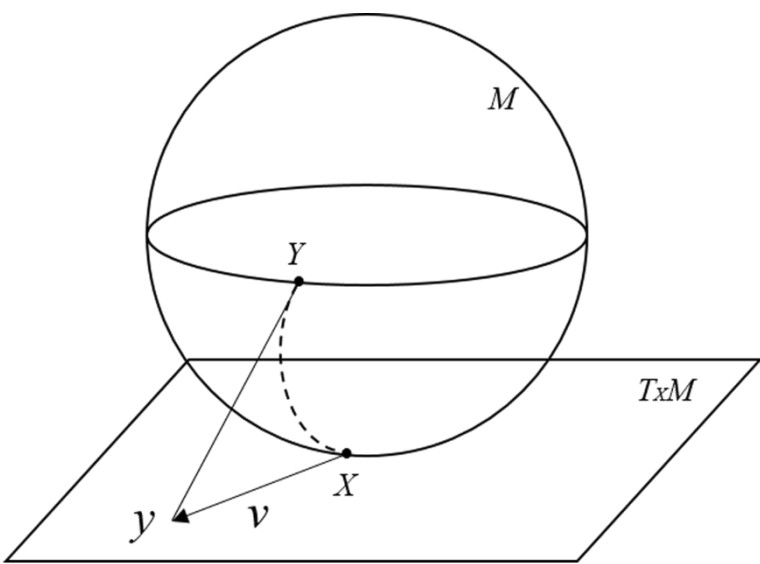

**Figure 5: Illustration of the tangent space $T_X M$ at point $X$ on a Riemannian manifold. A SPD matrix can be interpreted as point $X$ in the space of SPD matrices. The tangent vector $v$ can be obtained through the logarithm map, ie. $v = \log_X(Y)$. Every tangent vector in $T_X M$ can be mapped to the manifold through the exponential map, ie. $\exp_X(v) = Y$. The dotted line shows the geodesic starting at $X$ and ending at $Y$.**

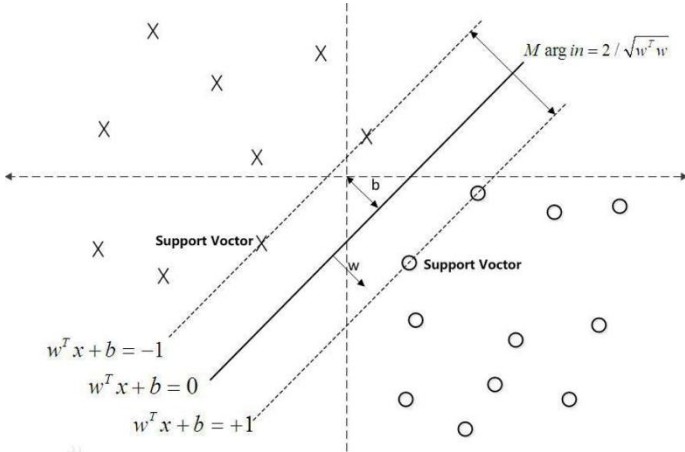

**Figure 6: The decision boundary of support vector machine with the largest margin. × and o denote two-class training examples, respectively. $w^T x + b = 0$ is the optimal hyperplane to do the separation, where $w$ is a weight vector and b is a bias, and an SVM training model with the largest margin $2/\sqrt{w^T w}$ is built. The support vectors are the samples on the dotted lines. The optimization classification hyperplane is determined by the solid line.**

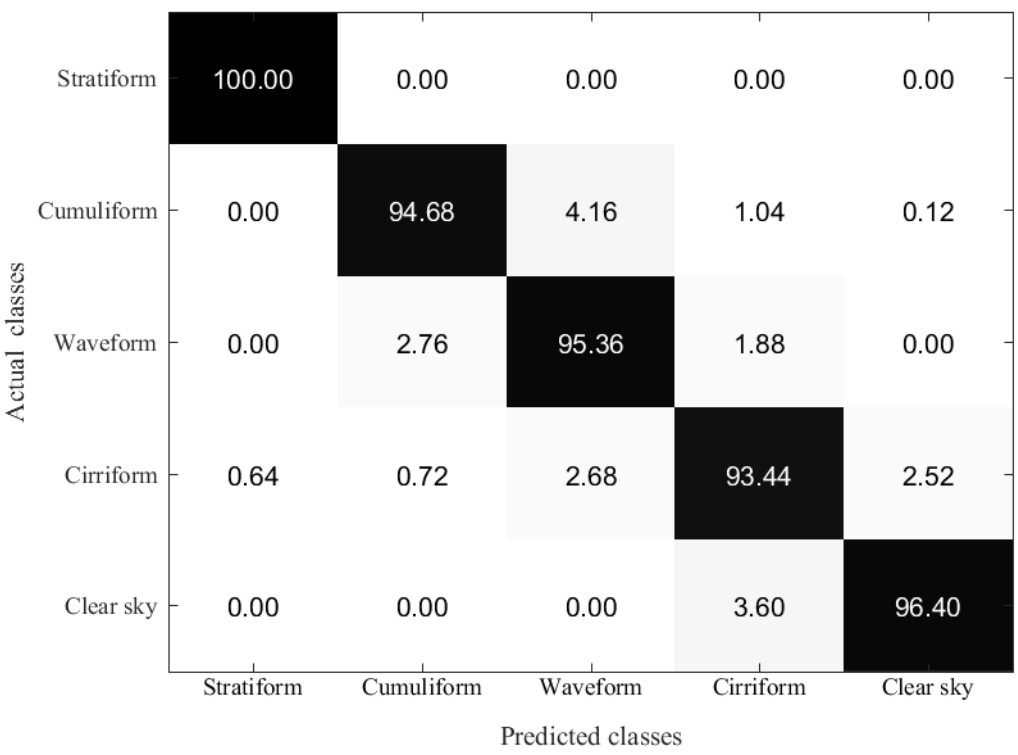

**Figure 7: Confusion matrix (%) on the zenithal dataset. (1/2 for training and the overall accuracy is 95.98%)**

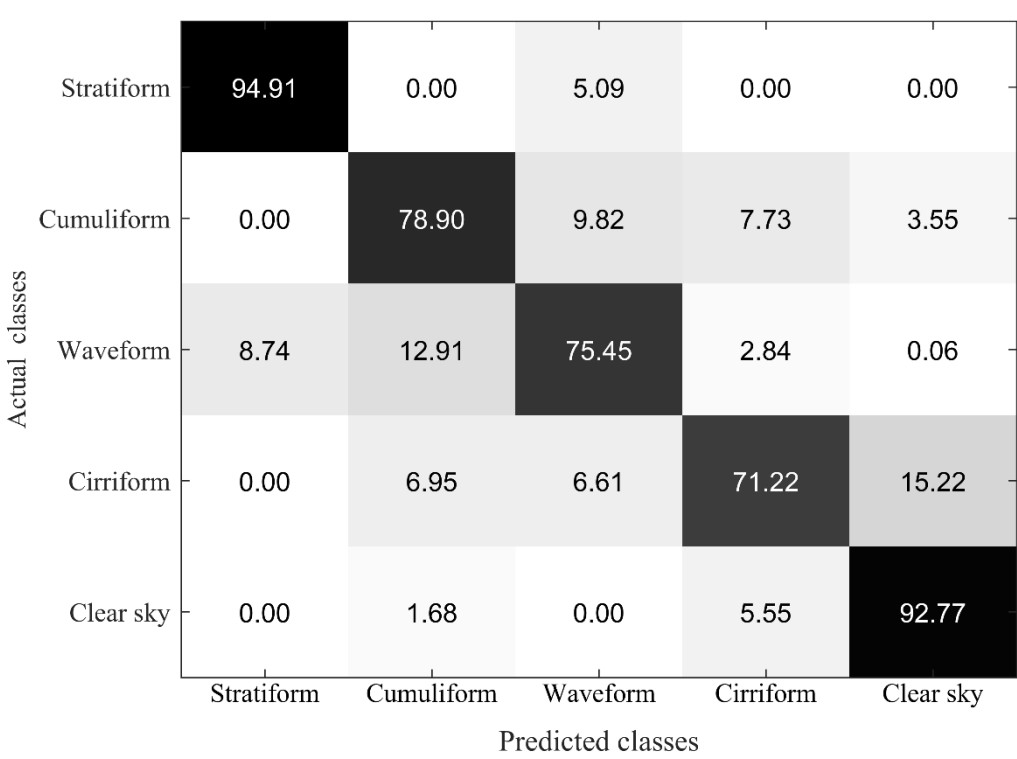

**Figure 8: Confusion matrix (%) on the whole-sky dataset. (1/2 for training and the overall accuracy is 83.54%)**

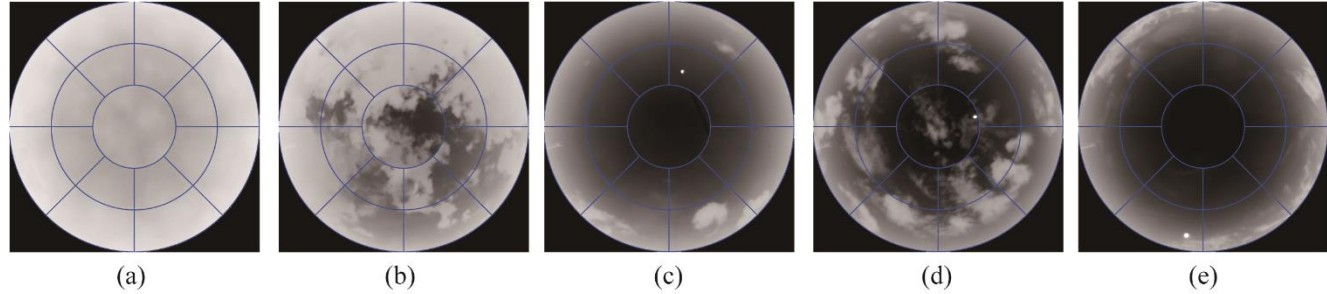

**Figure 9: Selected misclassified whole-sky images. (a) stratiform clouds to waveform clouds, (b) cumuliform clouds to waveform clouds, (c) cumuliform clouds to cirriform clouds, (d) waveform clouds to cumuliform clouds and (e) cirriform clouds to cumuliform clouds.**