# Peer review of "Cloud classification of ground-based infrared images combining manifold and texture features"

_Atmospheric Measurement Techniques, 2017_

## Referee Comment (RC1) · R. Clay (Referee) · 15 Feb 2018

The machine classification of cloud types found in automatically recorded images is an aim of considerable importance. However, it has proved difficult to develop suitable algorithms for this task. This paper combines two approaches, a texture analysis, such as one might expect based on statistical examination of the image structure, plus a manifold analysis such as is found effective, for instance, in facial recognition. The paper demonstrates that this combined process represents an improvement on previous analyses. The paper is interesting, well presented, and should be publishable.

The progress represented by this paper is incremental and the ultimate aim of classifying any cloud image is still distant. The paper demonstrates an ability to analyse

images of "high visual quality" and avoids "a complex mixture of cloud types" in its dataset, which is used both for training and analysis when manually analysed and then split into various groups. We are NOT addressing images with clouds of mixed types at various formation levels, which is a not-uncommon occurrence (noted as a next step in the Conclusions). However, the difficult issue of examining clouds away from the zenith, where the aspect of the cloud changes, is addressed with reasonable success.

The images under study are recorded in the long-wave infra-red. In these cases, the clear sky background brightness (temperature) varies with time and zenith angle (clear in figures 3 b,e). There is no discussion of whether those variations affect the image analysis, particularly the textural features which may have baseline issues. In the same sense, the camera (if radiometric) provides real information on the apparent temperature of the cloud, and this is unused.

So far as presentation is concerned, the paper is clearly written. Tables 4 and 5 should include some explanation of the fractions 1/10, 1 /2, 9/10 even though their meaning is clear from a reading of the text (minor revision).
* * *

---

## Author Comment (AC1) · 2 Mar 2018

We thank R. Clay for the insightful comments, which have allowed us to produce a stronger manuscript. Our responses to the comments are given below.

General comments:

The machine classification of cloud types found in automatically recorded images is an aim of considerable importance. However, it has proved difficult to develop suitable algorithms for this task. This paper combines two approaches, a texture analysis, such as one might expect based on statistical examination of the image structure, plus a manifold analysis such as is found effective, for instance, in facial recognition. The paper demonstrates that this combined process represents an improvement on previous analyses. The paper is interesting, well presented, and should be publishable.

The progress represented by this paper is incremental and the ultimate aim of classifying any cloud image is still distant. The paper demonstrates an ability to analyse images of "high visual quality" and avoids "a complex mixture of cloud types" in its dataset, which is used both for training and analysis when manually analysed and then split into various groups. We are NOT addressing images with clouds of mixed types at various formation levels, which is a not-uncommon occurrence (noted as a next step in the Conclusions). However, the difficult issue of examining clouds away from the zenith, where the aspect of the cloud changes, is addressed with reasonable success.

Specific comments:

1. Comment: The images under study are recorded in the long-wave infra-red. In these cases, the clear sky background brightness (temperature) varies with time and zenith angle (clear in figures 3 b, e). There is no discussion of whether those variations affect the image analysis, particularly the textural features which may have baseline issues.

**Response:** Many thanks for the thoughtful commenting. It is true that the clear sky background radiance in 8-14 μm varies with time and zenith angle. The images of the datasets have been preprocessed in the consideration of this important factor. The clear sky radiance threshold in each image is calculated using the radiation transfer model (Liu et al., 2013). The real radiance $R$ at each pixel in each image is converted to the grey value $G_{pixel}$ between [0,255] with $G_{pixel} = \frac{R}{R_{temp} - R_{clear}} * 255$, where $R_{clear}$ is the corresponding clear sky radiance threshold and $R_{temp}$ is the radiance corresponding to the real-time environment temperature. As a result, the effects of the clear sky background brightness temperature can be ignored, which means that this factor has little influence on the feature extraction of the images. In the revised manuscript, a further description of the cloud image preprocessing will be added in Section 2.1.

Reference

Liu, L., Sun, X., Gao, T., and Zhao, S.: Comparison of Cloud Properties from Ground-Based Infrared Cloud Measurement and Visual Observations, J. Atmos. Ocean. Tech., 30, 1171-1179, doi:10.1175/JTECH-D-12-00157.1, 2013.

2. Comment: In the same sense, the camera (if radiometric) provides real information on the apparent temperature of the cloud, and this is unused.

**Response:** Thanks for your constructive comments of the manuscript. The real information on the apparent temperature of the cloud is useful for analysing the images. The radiance value at each pixel in the image corresponds to a bright temperature. How to utilize the information of the brightness temperature effectively is worth further studying. At present, we mainly use the texture and manifold features to identify the cloud type. The addition of the brightness temperature of the physical information, or the combination of the height information obtained from the laser ceilometer might be helpful for the improvement of the cloud type recognition accuracy. We will carry out the research in the following work. Thanks again for the reviewer's advice. We will add this point to the section of Conclusions in the revised manuscript.

3. Comment: So far as presentation is concerned, the paper is clearly written. Tables 4 and 5 should include some explanation of the fractions 1/10, 1 /2, 9/10 even though their meaning is clear from a reading of the text (minor revision).

**Response:** Many thanks for your careful reading of our manuscript. We are sorry for our negligence. We will add some explanation of the fractions 1/10, 1 /2, 9/10 in Tables 4 and 5 in the revised manuscript to make it clear: 1/10, 1/2 and 9/10 are the certain proportions of the training set selected randomly from each category, and the rest part forms the testing set correspondingly.

---

## Referee Comment (RC2) · Anonymous Referee #2 · 16 Jul 2018

**Review for the article "Cloud Classification of ground-based infrared images combining manifold and texture features" by Qixiang Luo, Yong Meng, Lei Liu, Xiaofeng Zhao and Zeming Zhou**

In the article "Cloud Classification of ground-based infrared images combining manifold and texture features" the authors introduce a new method to identify cloud regimes from ground based cloud imagers. They base their study on a dataset from the Whole-Sky Infrared Cloud Measurement System, which provides zenith and whole sky images. The clouds in the images have been classified by two independent experts, and the image is only used in case they agree. The authors define a feature vector basing on the grey level co-occurrence matrix, which provides the measures energy, entropy, contrast and homogeneity; and on manifold features which are constructed by computing the regional covariance descriptor and mapping it into its tangent space. With their new method, the authors reach a slightly higher accurracy (by 3 to 5 %) compared to earlier methods of Liu and Cheng. The structure and the content of the paper are ok, but it lacks clarity in several places. Also, the language should be improved. I recommend major revisions. Please find my specific comments below:

**Major comments**

The article lacks some clarity. With regard to the underlying dataset, the authors explain how it is obtained and how the cases are chosen. However, in Section 3, "Experiments and discussion" the authors talk of "conducting each experiment 50 times on two datasets" (p.6, l.25). It is not clear to me what experiments are meant and in which way it can be repeated 50 times. Please state very clearly what you do and mean by this.

You mention a "Support Vector Machine" (SVM), which is used to perform the cloud classification. It is not clear what that actually is. Please extend the respective part a little bit, or give a citation at the very least.

The mathematical framework of a manifold is explained in quite detail. It distracts a little bit from the final result, the feature vector (supposedly Eq. 11). It would add much clarity to extend Sec. 223 "Combining manifold and texture features" and clearly state what you are now using for a cloud classification. You could also add information about the SVM here. Overall it is difficult to assess why this manifold features play a bigger role, or if for example other parameters would add equally much information. The mathematics behind the manifold feature vector is rather complicated, and a physical interpretation is hardly possible. Why has this vector been chosen? Would the addition of different, more easily physically interpretable parameters also lead to a higher hit rate in the classification? Is this more a "fitting problem" (more parameters → better fit) or is it more physically based? Please justify the choice of your metrics more.

The preselection of the data used in this study is done by employing two experienced experts, and the images are only chosen if both of them agree on the cloud type. Does that not already mean a very strong constraint on the images with regard to their clarity? Does it affect the study result? How would the algorithm perform under realistic conditions, where images are not preselected? Is there a quality flag involved? Is there a way to further improve the classification?

Please also give some insight how to assess your improvement of the classification. Depending on the case and fraction it is somewhere between 2 and 10% it seems. Is that a great improvement? Does it depend on the choice of cases?

**Minor comments**

Abstract: Overall, it already assumes a great background knowledge of the reader.

P1, l10: "the" Support Vector Machine → I think it is an overarching concept. "a" Support Vector Mache.

P1, l13: Specify some numbers here (higher by how much?)

P1, l16 and following: Somewhere you should mention and cite CloudNet (http://www.cloud-net.org) which are quite capable of identifying cloud types. Or do you only focus on large scale cloud structures? (The you should clarify that, because cloud classification implies that you look at the cloud type also.).

P1, l23,24: "weaken their credibility", please check the use of the word "credibility".

P1 l23 – P2, l13: This seems like an itemization of the existing methods. In which way do they connect to your method? Why do you later on chose just two of them (Liu, Cheng) to compare to?

P2, l5 "parallelepiped" → typo

P2, l10: Support Vector Machine needs a citation, it is not generally known.

P.2, l14 – l20: You state that colour images provide more information. Make clearer why infrared images are used anyway.

P.2, l21 – l24: Make it clear why manifolds are chosen. There are many mathematical constructs, it is not obvious why this way is chosen. Very clearly state here what the novelty and potential of your method is.

P.2, l 31: "...displayed the best performance in the 10-fold cross validation overall", it is not clear what 10-fold cross validation you mean.

P.3, l6: "...ground-based passive system that an uncooled microbolometer … is used." → "...ground-based passive system that uses an uncooled microbolometer …"

P.3, l 9: Are the pixels size or resolution?

P.3, l20: It is not clear why a historical dataset would not contain a complex mixture of cloud types compared to a dataset of the present.

P.3, l22: comprised of 100 images in each category

P.3, l26: the number of cases with stratiform clouds, cumuliform clouds, …

P.3, l29: "which is the area of clouds rather than the parts out of the circle" → what do you mean by "parts out of the circle"?

P.4, l5: Only later clear what "non-Euclidean features" are

P.4, l22: "...mean values in four directions are obtained as texture feature", mean over what? And what directions?

P.5, l10: This is supposedly a d times d matrix. Should the d not show up in this equation as an index or something?

P.6, l3: This is not an equation, it lacks a left side. P.6, Section 2.3: Needs some more explanation or at the least citations. SVM not understandable from this.

P.6, l.21: What do you mean by "voting policy"?

P.6, Section 3, beginning: Here, it should be clarified at the very latest what you mean by "experiment" in your context.

P.7, l21: Do you really mean "confusion matrix"?

P.7, l24: cululiform → cumuliform

P.7, l27: "has reached" → "is reached"

P.7, l30: exits → exists

P.8, l4: "when 1/2 for training." → "when 1/2 for training is used."

P.8, l25: There is indeed improvement, but I would not call it "dramatically".

P.8, l26: What do you mean by "the statistical learning method"? I think this hasn't been defined before.

P.8, l32: Gabor or wavelet coefficients may need a citation, not generally known.

P.9, l3: "on the both" → "on both,"

Images and Tables:

Please provide more telling captions.

Table 4 and 5: The 1/10, 1/2 and so on are not clear

---

## Author Comment (AC2) · 9 Aug 2018

We thank the anonymous reviewer for the insightful and thoughtful comments, which have allowed us to produce a stronger manuscript. Our responses to the comments are given below, and the corresponding changes are marked in red at specific location **(Page X, Line X)** in the revised manuscript.

**General comments:**

In the article "Cloud Classification of ground-based infrared images combining manifold and texture features" the authors introduce a new method to identify cloud regimes from ground based cloud imagers. They base their study on a dataset from the Whole-Sky Infrared Cloud Measurement System, which provides zenith and whole sky images. The clouds in the images have been classified by two independent experts, and the image is only used in case they agree. The authors define a feature vector basing on the grey level co-occurrence matrix, which provides the measures energy, entropy, contrast and homogeneity; and on manifold features which are constructed by computing the regional covariance descriptor and mapping it into its tangent space. With their new method, the authors reach a slightly higher accuracy (by 3 to 5 %) compared to earlier methods of Liu and Cheng. The structure and the content of the paper are ok, but it lacks clarity in several places. Also, the language should be improved. I recommend major revisions. Please find my specific comments below.

**Response:** Many thanks for the constructive and valuable comments of the manuscript. We have tried our best to enrich the paper's content to make it clearer. Besides, we have polished the language as far as we can to satisfy the requirement of publication.

**Specific comments:**

1. Comment: The article lacks some clarity. With regard to the underlying dataset, the authors explain how it is obtained and how the cases are chosen. However, in Section 3, "Experiments and discussion" the authors talk of "conducting each experiment 50 times on two datasets" (p.6, l.25). It is not clear to me what experiments are meant and in which way it can be repeated 50 times. Please state very clearly what you do and mean by this.

**Response: (Page 8, Line 13-18)** We are sorry not to make it clear. In Section 3, "Experiments and discussion" validates which features are chosen, presents the results and gives the discussion. We first adopt 10-fold cross validation to determine which features among texture features, manifold features and combined features perform best. 10-fold cross validation means that each dataset is divided into 10 subsets with the same size at random in turn, then one single subset is used for validation and the other 9 parts are taken as the training set. To test the performance of the algorithm, 10-fold cross validation is conducted 50 times and average values are taken as final results.

 The combined features are used for the cloud type recognition experiments. Different from a deterministic case, the training samples of the experiments are chosen randomly. The purpose that we conduct the experiments is to test the performance of the algorithm. In each experiment, the training set

is selected at random and the rest of the dataset forms the testing set, so we need to repeat this process many times and calculate the mean value in order to reduce the accidental bias and to measure the algorithm well.

2. Comment: You mention a "Support Vector Machine" (SVM), which is used to perform the cloud classification. It is not clear what that actually is. Please extend the respective part a little bit, or give a citation at the very least.

**Response: (Page 7, Line 23-Page 8, Line 5)** Many thanks for the constructive comments of the manuscript. In the area of machine learning, Support Vector Machines (SVMs) are supervised learning models (Cristianini and Shawe-Taylor, 2000). An SVM model is a representation of the examples as points in the Reproducing Kernel Hilbert Space, mapped so that the examples of the separate categories are divided by a clear gap that is as wide as possible. New examples are then mapped into that same space and predicted to belong to a category based on which side of the gap they fall.

As Fig.1 shows, given a set of two-class training examples (denoted by × and o, respectively), the key problem is to find the optimal hyperplane to do the separation: $w^T x + b = 0$, where $w$ is a weight vector and $b$ is a bias, and an SVM training model with the largest margin $(2/\sqrt{w^T w})$ is built. The support vectors are the samples on the dotted lines. The optimization classification hyperplane is determined by the solid line. The test examples are assigned to one category or the other based on this model, making it a non-probabilistic binary linear classifier. In this work, we apply a simple linear function as the mapping kernel.

[Figure]

Figure 1: The decision boundary of support vector machine with the largest margin. × and o denote two-class training examples, respectively. $w^T x + b = 0$ is the optimal hyperplane to do the separation, where $w$ is a weight vector and $b$ is a bias, and an SVM training model with the largest margin $2/\sqrt{w^T w}$ is built. The support vectors are the samples on the dotted lines. The optimization classification hyperplane is determined by the solid line.

3. Comment:
1) The mathematical framework of a manifold is explained in quite detail. It distracts a little bit from the final result, the feature vector (supposedly Eq. 11).
2) It would add much clarity to extend Sec. 2.2.3 "Combining manifold and texture features" and clearly state what you are now using for a cloud classification.

3) You could also add information about the SVM here.

4) Overall it is difficult to assess why this manifold features play a bigger role, or if for example other parameters would add equally much information.

5) The mathematics behind the manifold feature vector is rather complicated, and a physical interpretation is hardly possible. Why has this vector been chosen?

6) Would the addition of different, more easily physically interpretable parameters also lead to a higher hit rate in the classification?

7) Is this more a "fitting problem" (more parameters → better fit) or is it more physically based? Please justify the choice of your metrics more.

**Response:** Many thanks for the constructive comments of the manuscript.

1) **(Page 6, Line 1-Page 7, Line 15)** Generally speaking, the manifold is a topological space that is locally equivalent to a Euclidean space. The differential manifold has a globally defined differential structure. Its tangent space $T_X M$ is a space formed by all possible tangent vectors at a given point $X$ on the differential manifold. For the Riemannian manifold $M$, an inner product is defined in its tangent space. The shortest curve between two points on the manifold is called a geodesic and the length of the geodesic is the shortest distance between two points.

All SPD matrices form a Riemannian manifold. Suppose $S^d$ is a set of all $n \times n$ real symmetric matrices: $S^d = \{A \in M(d): A^T = A\}$, where $M(d)$ represents the set of all $d \times d$ matrices, so that $S^d_{++} = \{A \in S^d: A > 0\}$ is the set of all $d \times d$ SPD matrices, which construct a $d(d + 1)/2$ dimensional SPD manifold. According to the operation rules of the matrix, $S^d$ is a vector space while $S^d_{++}$ is a non-Euclidean space. A Riemannian metric should be given to describe the geometric structure of the SPD matrix and to measure the distance of two points on $S^d_{++}$.

Geodesics on the manifold are related to the tangent vectors in the tangent space. Two operators, namely exponential map $\exp_X(\cdot): T_X M \to M$ and the logarithm map $\log_X(\cdot) = \exp_X^{-1}(\cdot): M \to T_X M$, are defined over differentiable manifolds to switch between the manifold and tangent space at the point $X$. As illustrated in Fig. 2, the tangent vector $v$ is mapped to the point $Y$ on the manifold through the exponential map. The length of v is equivalent to the geodesic distance between X and Y based on the property of the exponential map. The logarithm map is the inverse of the exponential map and maps a point on the manifold to the tangent space $T_X M$. Conversely, a point on the manifold is mapped to the tangent space $T_X M$ through the logarithm map. The exponential and logarithm maps vary as point X moves along the manifold. The details can be referred in Harandi et al. (2012).

[Figure]

**Figure 2: Illustration of the tangent space $T_XM$ at point $X$ on a Riemannian manifold. A SPD matrix can be interpreted as point $X$ in the space of SPD matrices. The tangent vector $v$ can be obtained through the logarithm map, ie. $v = \log_X(Y)$. Every tangent vector in $T_XM$ can be mapped to the manifold through the exponential map, ie. $\exp_X(v) = Y$. The dotted line shows the geodesic starting at $X$ and ending at $Y$.**

For $S_{++}^d$, the logarithm and exponential maps are defined as:

$$\log_X(Y) = X^{\frac{1}{2}}\log(X^{-\frac{1}{2}}YX^{-\frac{1}{2}})X^{\frac{1}{2}}, \tag{1}$$

$$\exp_X(y) = X^{\frac{1}{2}}\exp(X^{-\frac{1}{2}}yX^{-\frac{1}{2}})X^{\frac{1}{2}}, \tag{2}$$

where $\log(\cdot)$ and $\exp(\cdot)$ are the matrix logarithm and exponential operators, respectively. For SPD matrices, they can be computed through Singular Value Decomposition (SVD). If we let $\text{diag}(\lambda_1, \lambda_2, ..., \lambda_d)$ be a diagonal matrix formed from real values $\lambda_1, \lambda_2, ..., \lambda_d$ on diagonal elements and $X = U\text{diag}(\lambda_i)U^T$ be the SVD of the symmetric matrix $X$. In Eq. (1) and Eq. (2), $\log(\cdot)$ and $\exp(\cdot)$ are calculated by

$$\log(X) = \sum_{r=1}^{\infty}\frac{(-1)^{r-1}}{r}(X - I)^r = U\text{diag}(\ln(\lambda_i))U^T, \tag{3}$$

$$\exp(X) = \sum_{r=0}^{\infty}\frac{1}{r!}X^r = U\text{diag}(\exp(\lambda_i))U^T, \tag{4}$$

where $I$ is an identity matrix on manifolds.

The manifold can be embedded into its tangent space at identity matrix $I$. Thus, based on the bi-invariant Riemannian metric (Arsigny et al., 2008), the distance between two SPD matrices $X$, $Y$ is $d(X, Y) = \|\log(X) - \log(Y)\|_2$, where $\log(\cdot)$ is the matrix logarithm operator. Since symmetric matrices (equivalently tangent spaces) form a vector space, the classification tools in the Euclidean space (SVM, KNN and so on) can be seamlessly employed to deal with the recognition problem.

Given an SPD matrix $A$, its log-Euclidean vector representation $a \in \mathbb{R}^m$, $m = d(d + 1)/2$, is unique and defined as $a = Vec(\log(A))$. Let $B = \log(A)$, $B \in S^d$ and

$$B = \begin{bmatrix} b_{1,1} & b_{1,2} & b_{1,3} & ... & b_{1,d} \\ b_{2,1} & b_{2,2} & b_{2,3} & ... & b_{2,d} \\ \vdots & \vdots & \vdots & ... & \vdots \\ b_{d,1} & b_{d,2} & b_{d,3} & ... & b_{d,d} \end{bmatrix}_{d \times d}, \tag{5}$$

which lies in the Euclidean space. Since $B$ is symmetric, we can rearrange it into a vector by vectorizing its upper triangular matrix:

$$a = Vec(B) = \begin{bmatrix} b_{1,1}, \sqrt{2}b_{1,2}, \cdots, \sqrt{2}b_{1,d}, b_{2,2}, \sqrt{2}b_{2,3}, \cdots, b_{d,d} \end{bmatrix}^{\mathrm{T}}. \tag{6}$$

Thus, vector a is defined as the manifold features. Since $f$ is a 6-dimensional feature mapping in the experiment, the manifold feature vector $a$ to represent the cloud image is $6 \times (6 + 1)/2 = 21$ dimensions. The mapped feature vector can reflect the characteristics of its corresponding SPD matrix on matrix manifolds. Thus, manifold features can describe the non-Euclidean property of the infrared image features to some degree.

2) **(Page 7, Line 16-21)** Section 2.2.3 Combining manifold and texture features

As described in Sect. 2.2.1 and 2.2.2, manifold features and texture features can be extracted and integrated to represent the ground-based infrared images. For an image, the four features including energy, entropy, contrast and homogeneity from GLCM, express its texture, while 21-demensional manifold features describe the non-Euclidean geometric characteristics. The manifold and texture features are combined to form a feature vector to represent an image. Thus, the joint features of the infrared image have a total of 25 dimensions.

3) **(Page 7, Line 23-Page 8, Line 5)** The information of SVM has be added in Section 2.3 Classification.

4) In the experiment, as shown in Table 1 (Corresponding to Table 3 in the manuscript), we first test the effects of different features. It is shown that manifold features perform better than texture features with an increase of at least 10% on the zenithal dataset and about 3.5% on the whole-sky dataset. When these two features are combined, there is an improvement on the 10-fold cross validated classification accuracy compared to using texture or manifold features alone. It's clear on the whole-sky dataset, the accuracy is improved by about 2.7%. Comparing two features alone, manifold features provide more discriminate information than texture features. As a result, manifold features play a bigger role in the combined features in our experiment.

**Table 1. The 10-fold cross validated classification accuracy (%) of the proposed method on two datasets.**

|  | Zenithal | Whole-sky |
|---|---|---|
| Texture features | 83.49 | 78.01 |
| Manifold features | 96.46 | 82.38 |
| Combined features | **96.50** | **85.12** |

5) **(Page 2, Line 21-31)** In this paper, we utilize region covariance matrices, composed from densely sampled features, as the descriptors. This region descriptor has several advantages. Firstly, it calculates the first-order and second-order statistics of the local patch. Secondly, it straightforward fuses various

features. Thirdly, it is independent of the region size and has low dimensions. Fourthly, by subtracting the mean feature vector, the effect of the noisy samples is reduced to some degree. Finally, it is able to speed up the computation in images and videos using efficient methods (Tuzel et al., 2008; Sanin et al., 2013). Covariance matrices are Symmetric Positive Definite (SPD) matrices and naturally form a connected Riemannian manifold. The manifold feature vector can maintain these advantages and describe the images well, so it is chosen for a try on the ground-based cloud classification.

6)    In our work, we have tested other features proposed in the literatures. For example, 6-dimensional features including mean, standard deviation, smoothness, third moment, uniformity and entropy in Calbó and Sabburg (2008) and 4-dimensional features including mean, standard deviation, skewness and cloud cover in Heinle et al. (2010) were added with texture features for classification of ground-based cloud types. Table 2 gives the 10-fold cross validation results using these features compared to the proposed method on the zenithal dataset and whole-sky dataset, respectively.

In Table 2, 10-dimensional features are the combination of 4-dimensional texture features, energy, entropy, contrast and homogeneity of GLCM and 6-dimensional features used in Calbó and Sabburg (2008), 8-dimensional features are the combination of these 4-dimensional texture features from GLCM and 4-dimensional features adopted in Heinle et al. (2010) and 25-dimensional features are the proposed method in this manuscript. As shown in Table 2, the results of 25-dimensional features outperform those of the other two combined features. It means that on these two datasets, combination of manifold features and texture features performs better. Other different, more easily physically interpretable parameters may exist and lead to a higher hit rate in the classification. In our manuscript, the addition of manifold features has validated its effectiveness in the ground-based cloud classification on the two datasets.

**Table 2. The 10-fold cross validated classification accuracy (%) with different features on two datasets.**

|  | Zenithal | Whole-sky |
| --- | --- | --- |
| 10-dimenisional features | 94.76 | 82.59 |
| 8-dimenisional features | 92.91 | 82.01 |
| 25-dimensional features | **96.50** | **85.12** |

7)    Essentially, cloud classification is not a physically based problem, but a problem of pattern recognition. The classification accuracy is mainly dependent on the features and the classifier. Features are used to represent the image itself, and the classifier acts as a referee to predict its type according to its features. For example, if we can justify all the images based on one feature, there is no need to adopt the others. The best features should represent the most significant properties of the images well. That's not to mean that the more parameters, the better fit.

4. Comment: The preselection of the data used in this study is done by employing two experienced experts, and the images are only chosen if both of them agree on the cloud type. Does that not already mean a very strong constraint on the images with regard to their clarity? Does it affect the study result? How would the algorithm perform under realistic conditions, where images are not preselected? Is there a quality flag involved? Is there a way to further improve the classification?

**Response: (Page 3, Line 26–Page 4, Line 6)** Many thanks for the constructive comments of the

manuscript. It is true that the preselection of the data used in this study is done by employing two experienced experts, and the images are only chosen if both of them agree on the cloud type. The sentence "The selection criterion is that the chosen images should hold high visual quality and can be recognized by visual inspection" has been changed to "The selection premise is that …". The purpose that the images should hold high visual quality is to make them recognized by visual inspection accurately. If an image is vague, it's hard for experts to justify its type. For the algorithm, it's hard to extract effective features of a vague image, not to mention recognizing its cloud type. In general, the ground-based images are obtained by the specific equipment, like Whole-Sky Infrared Cloud-Measuring System (WSIRCMS), so the quality of the images is similar in most cases. There is no specific quality flag to select images, and it is mainly dependent on the visual judgment. Under realistic conditions, according to the previous data gathered in one place, the SVM models are trained to predict the real-time images there. The results given by automatic cloud classification algorithm are compared with those determined by experts to evaluate the performance of the algorithm. To further improve the classification accuracy, we should gather as many representative samples in per class as possible. When the training images are typical and sufficient enough, there is no doubt that the recognition rate will be improved.

5. Comment: Please also give some insight how to assess your improvement of the classification. Depending on the case and fraction it is somewhere between 2 and 10% it seems. Is that a great improvement? Does it depend on the choice of cases?

**Response: (Page 9, Line 10-14; Page 9, Line 31-Page 10, Line 2)** Many thanks for the careful reading of our manuscript. In the manuscript, we give three cases to make comparison with different methods. First, when 1/10 of the dataset is treated as the training set, there is an increase of about 9.2% for the zenithal dataset and about 4.7% for the whole-sky dataset in the overall accuracy. This is a case that the training samples are not sufficient. Second, we conduct the experiment when 1/2 of the dataset is used for training. There is an increase of about 3.7% for the zenithal dataset and about 3% for the whole-sky dataset in the overall accuracy. This is a case that the training samples are enough. Third, we conduct the experiment when 9/10 of the dataset is used for training. There is an increase of about 2.9% for the zenithal dataset and about 3.7% for the whole-sky dataset in the overall accuracy. This is a case that the training samples are quite enough. With different fractions that the training set occupies, it is validated that in most cases the proposed method outperforms the other two methods (Liu et al., 2015; Cheng and Yu, 2015). As a whole, the improvement of the proposed method is between 2% and 10%. To some degree, it may not be a great improvement, but we have validated that the introduction of manifold features is effective and can achieve some success, it is worthy doing more work in this field to promote its development. In general, with the increase in the number of training samples, the overall accuracy will increase until it holds stable.

6. Comment: Abstract: Overall, it already assumes a great background knowledge of the reader.

**Response:** Many thanks for the constructive comments of our manuscript. We have added some introduction of SVM and other explanations in the revised manuscript.

7. Comment: P1, l10: "the" Support Vector Machine → I think it is an overarching concept. "a" Support Vector Mache.

**Response: (Page 1, Line 10)** Many thanks for the thoughtful commenting. We agree and have changed in the revised manuscript.

8. Comment: P1, l13: Specify some numbers here (higher by how much?)

**Response: (Page 1, Line 13)** Many thanks for the thoughtful comment. The proposed method performs higher by 2%-10% than the other two methods in the recognition rate.

9. Comment: P1, l16 and following: Somewhere you should mention and cite CloudNet (http://www.cloud-net.org) which are quite capable of identifying cloud types. Or do you only focus on large scale cloud structures? (The you should clarify that, because cloud classification implies that you look at the cloud type also.).

**Response: (Page 1, Line 19-21)** Many thanks for the constructive comments of the manuscript. The categorization products at the CloudNet Web site (http://www.cloud-net.org) form a dataset, aggregated from cloud radar, lidar, a numerical forecast model and optionally a rain gauge and microwave radiometer. They are used to improve the numerical prediction models. The observational datasets and model datasets are used in this procedure. Different from it, we focus on identifying cloud types based on the images, so it is a problem to extract features due to the formats of the input data. Compared to the traditional manual observation, ground-based infrared cloud images can be obtained continuously and have a high spatial resolution at a local scale. Also, the title "Cloud Classification of **ground-based infrared images** combining manifold and texture features" has clarified our purpose. We have added Illingworth et al. (2007) as a reference in the revised manuscript.

10. Comment: P1, l23,24: "weaken their credibility", please check the use of the word "credibility".

**Response: (Page 1, Line 24)** Many thanks for the careful reading. We have checked the use of the word "credibility", and the original sentence (the last sentence in the first paragraph of Section Introduction) in Tzoumanikas et al. (2012) is "Additionally, the introduction of human factor and a rough measuring system (in octas or tenths of the sky dome) in the estimation of cloud cover and type weaken their credibility.", so we use this word in the manuscript.

11. Comment: P1 l23 – P2, l13: This seems like an itemization of the existing methods. In which way do they connect to your method? Why do you later on choose just two of them (Liu, Cheng) to compare to?

**Response:** Many thanks for the constructive comments. As mentioned in the manuscript, there are some methods for ground-based cloud classification. Usually, the main steps contain feature extraction and cloud type recognition with the classifier. In the manuscript, we stated what features and classifiers were adopted in the literatures. As a result, we should hand these two sub-problems well to realize our goal. In these existing methods, most features including statistical, texture, and structure features are extracted in the Euclidean space. Different from these methods, besides the texture features, manifold features are extracted on Riemannian manifolds to describe non-Euclidean geometric characteristics in our method.

Most methods are conducted on ground-based colour images, and the extracted features need

different channels' information. Infrared images only have one channel, so it's hard to realize these methods based on colour features. Liu's method uses Weighted Local Binary Pattern (WLBP), which just based on the grayscale images (Liu et al., 2015). Cheng's method uses statistical features and distribution of local texture features (Cheng and Yu, 2015). Differences of R-G, R-B and G-B components are given up to make Cheng's method apply to the infrared images. Then these two methods are adopted for comparison.

12. Comment: P2, l5 "parallelepiped" → typo

**Response: (Page 2, Line 5)** Many thanks for the careful reading. We have checked the spelling of the word "parallelepiped", and the original sentence (the first sentence in the fourth paragraph of Section Results and discussion) in Calbó and Sabburg (2008) is "The classifier that we developed is based on the supervised **parallelepiped** technique, which has been used elsewhere for similar applications", so we adopt this terminology in our manuscript.

13. Comment: P2, l10: Support Vector Machine needs a citation, it is not generally known.

**Response: (Page 2, Line 11; Page 7, Line 24)** Many thanks for the careful reading of our manuscript. We have added Cristianini and Shawe-Taylor (2000) as a reference to make it clear.

14. Comment: P.2, l14 – l20: You state that colour images provide more information. Make clearer why infrared images are used anyway.

**Response: (Page 2, Line 16-19)** Many thanks for the careful reading of our manuscript. It is known that the colour image has 3 channels of R, G and B while the infrared image only has one channel. As a result, from the aspect of the image itself, the RGB images can provide colour information, which is important for analyzing the cloud to some degree. Although colour images have this advantage, it is hard to gather them during the night. Compared to colour images, infrared images can be obtained day and night continuously, which is necessary for practical application. As a result, we investigate the ground-based infrared images.

15. Comment: P.2, l21 – l24: Make it clear why manifolds are chosen. There are many mathematical constructs, it is not obvious why this way is chosen. Very clearly state here what the novelty and potential of your method is.

**Response: (Page 2, Line 21-31)** We are sorry for our inadequate explanation. Nowadays, the Symmetric Positive Definite (SPD) matrix manifold has achieved success in many aspects, such as action recognition, material classification and image segmentation (Faraki et al., 2015; Jayasumana et al., 2015). It is because Euclidean space cannot gain special structures of image features well while Riemannian manifolds form non-Euclidean spaces and are more appropriate to address this problem. Although it proves effective, few researches are pursued for the task of cloud classification with manifold features.

In this paper, we utilize region covariance matrices, composed from densely sampled features, as the descriptors. A region descriptor has several advantages. Firstly, it calculates the first-order and second-order statistics of the local patch. Secondly, it straightforward fuses various features. Thirdly, it

is independent of the region size and has low dimensions. Fourthly, by subtracting the mean feature vector, the effect of the noisy samples is reduced to some degree. Finally, it is able to speed up the computation in images and videos using efficient methods (Tuzel et al., 2008; Sanin et al., 2013). Covariance matrices are SPD matrices and naturally form a connected Riemannian manifold. Covariance matrices are Symmetric Positive Definite (SPD) matrices and naturally form a connected Riemannian manifold. Although it proves effective, few researches are pursued for the task of cloud classification with manifold features. The manifold feature vector can maintain these advantages and describe the images well, so it is chosen for a try on the cloud classification. In this paper, a novel cloud classification method is proposed for ground-based infrared images. Manifold features, representing the non-Euclidean geometric structure of the image features, and texture features, expressing the image texture, are integrated for the feature extraction.

16. Comment: P.2, l 31: "...displayed the best performance in the 10-fold cross validation overall", it is not clear what 10-fold cross validation you mean.

**Response: (Page 3, Line 5)** We are sorry for our inadequate explanation. Cross validation is a usual way to test the performance of the algorithm (Ripley, 2005), which has been applied by e.g. Heinle et al. (2010), Li et al. (2016) and Gan et al. (2017). In the 10-fold cross validation, the dataset is divided into 10 subsets with the same size at random, then one single subset is used for validation in turn and the other 9 parts are taken as the training set. This process is then repeated 10 times. The average value is used to estimate the capability of the algorithm. In general, 10-fold cross validation is repeated many times to measure the algorithm's performance. We have added Ripley (2005) as a reference and clarified it later (**Page 8, Line 16-18**) in the revised manuscript.

17. Comment: P.3, l6: "...ground-based passive system that an uncooled microbolometer ⋯ is used." → "...ground-based passive system that uses an uncooled microbolometer …"

**Response: (Page 3, Line 14)** Many thanks for the careful reading of our manuscript. We have made this change in the revised manuscript.

18. Comment: P.3, l 9: Are the pixels size or resolution?

**Response: (Page 3, Line 17)** Sorry to make it confusing. Here are the pixels' size, not the resolution.

19. Comment: P.3, l20: It is not clear why a historical dataset would not contain a complex mixture of cloud types compared to a dataset of the present.

**Response: (Page 4, Line 4-5)** We are sorry for our inadequate explanation. The dataset is used to assess the performance of the algorithm. The sky condition is classified into five types: stratiform clouds, cumuliform clouds, waveform clouds, cirriform clouds and clear sky. To guarantee the reliability of true label of each image, we should select the images without mixed cloud types. Indeed, the sky condition with mixed cloud types always exists, but it is complicated. We will investigate the images containing mixed cloud types next.

20. Comment: P.3, l22: comprised of 100 images in each category

**Response: (Page 4, Line 6)** Many thanks for the careful reading of our manuscript. We have made this change in the revised manuscript.

21. Comment: P.3, l26: the number of cases with stratiform clouds, cumuliform clouds, …

**Response: (Page 4, Line 11)** We are sorry for our carelessness. We have made this change in the revised manuscript.

22. Comment: P.3, l29: "which is the area of clouds rather than the parts out of the circle" → what do you mean by "parts out of the circle"?

**Response: (Page 4, Line 13-15; Page 22)** We are sorry to make it confusing. In Fig. 3, it can be seen that the region of interest is the white region while the black region is out of the circle.

[Figure]

**Figure 3: The mask of the whole-sky images.**

23. Comment: P.4, l5: Only later clear what "non-Euclidean features" are.

**Response: (Page 7, Line 18-20)** Many thanks for the constructive comment. In the manuscript, the manifold features describe the non-Euclidean geometric characteristics of infrared image features. As each image corresponds to a SPD matrix, to better maintain its manifold geometric structure, it is mapped into its tangent space by the logarithm operation. The mapped feature vector can reflect the characteristics of its corresponding SPD matrix on matrix manifolds. Thus, manifold features can describe the non-Euclidean property of the infrared image features to some degree.

24. Comment: P.4, l22: "...mean values in four directions are obtained as texture feature", mean over what? And what directions?

**Response: (Page 5, Line 6-10)** We are sorry not to make it clear. The Grey Level Co-occurrence Matrix (GLCM) is calculated in a defined direction θ and a pixel distance d. In the experiment, we get four GLCMs with d=1 and θ=0°, 45°, 90°, 135°. Then four measures energy, entropy, contrast and homogeneity are computed according to Eq. (7) ~ Eq. (10). Since there are 4 GLCMs, the texture features are 16 dimensional. To alleviate the complexity, reduce the dimension and keep rotation invariance, four mean features of four GLCMs with d=1 and θ=0°, 45°, 90°, 135°are obtained as the final texture features.

$$\text{Energy} = \sum_{i=0}^{k-1} \sum_{j=0}^{k-1} p(i,j)^2. \tag{7}$$

$$\text{Entropy} = -\sum_{i=0}^{k-1} \sum_{j=0}^{k-1} p(i,j) \log_2 p(i,j). \tag{8}$$

$$\text{Contrast} = \sum_{i=0}^{k-1} \sum_{j=0}^{k-1} (i-j)^2 p(i,j)^2. \tag{9}$$

$$\text{Homogeneity} = \sum_{i=0}^{k-1} \sum_{j=0}^{k-1} \frac{p(i,j)}{1+|i-j|}. \tag{10}$$

25. Comment: P.5, l10: This is supposedly a d times d matrix. Should the d not show up in this equation as an index or something?

**Response: (Page 5, Line 19)** Many thanks for the constructive comments. We have checked this equation and there is no mistake for this equation. It is in the form of matrix operation. For a feature image $F$, it contains n=W×H points of d-dimensional features $\{f_k, k=1, 2, ..., n\}$. Its Covariance Descriptor (CovD) is a d×d covariance matrix, computed by

$$C = \frac{1}{n-1} \sum_{k=1}^{n} (f_k - \mu)(f_k - \mu)^{\mathrm{T}}, \tag{11}$$

where $\mu = \frac{1}{n} \sum_{k=1}^{n} f_k$, which represents the feature mean vector. In this equation, $f_k$ is a d-dimensional feature vector, and as the feature mean vector, $\mu$ is also d-dimensional, so $(f_k - \mu)(f_k - \mu)^{\mathrm{T}}$ is a d×d matrix, and the CovD $C$ is a d×d matrix. The $(i, j)$-th element of CovD in Eq. (11), $C(i,j)$, can also written as:

$$C(i,j) = \frac{1}{n-1} \sum_{k=1}^{n} \left( f_k(i) - \mu(i) \right)\left( f_k(j) - \mu(j) \right), \quad i,j = 1,2,...,d, \tag{12}$$

where $f_k(i)$, $\mu(i)$ mean the $i$-th elements of the feature vectors $f_k$ and $\mu$, respectively.

26. Comment: P.6, l3: This is not an equation, it lacks a left side. P.6, Section 2.3: Needs some more explanation or at the least citations. SVM not understandable from this.

**Response: (Page 7, Line 4)** Many thanks for the careful reading. This equation has been modified as follows:

$$A^* = \arg \min_A \|C - A\|_F, \text{s.t.} A + A^T > 0. \tag{13}$$

27. Comment: P.6, l.21: What do you mean by "voting policy"?

**Response: (Page 8, Line 7-11)** Many thanks for the thoughtful comments. In a multi-class task, the SVM is conducted between every two classes. In this paper, there are 5 types, so there are 5×(5-1)/2=10 SVM classifiers between every two classes. For an unknown-type image, it will be input into 10 models and get 10 output labels. That is to say, each binary classifier makes its vote to predict the sample's class. According to the voting policy, the most frequent label is this sample's type.

28. Comment: P.6, Section 3, beginning: Here, it should be clarified at the very latest what you mean by "experiment" in your context.

**Response: (Page 8, Line 12-16)** Many thanks for the constructive comment. Different from a concrete case, the experiment we carry out is random. The purpose that we conduct the experiment is to test the performance of the algorithm. In the experiment, the training set is selected at random and the rest of the dataset forms the testing set, so we need to repeat this process many times and calculate the mean value in order to avoid the accidental bias and to measure the algorithm well. In the revised manuscript, we have checked the "experiment" and "cross validation" to make it clear.

29. Comment: P.7, l21: Do you really mean "confusion matrix"?

**Response: (Page 9, Line 17-19)** Many thanks for the careful comment. The confusion matrix is a way to exhibit the experimental result, which has been applied by e.g. Zhuo et al. (2014), Liu et al. (2015), and Li et al. (2016). In the confusion matrix, each row of the matrix represents an actual class while each column represents the predicted class given by the algorithm. For example, the element in the second row and third column is the percentage of cumuliform clouds misclassified as waveform clouds. Therefore, the recognition rate for each class is in the diagonal of the matrix.

30. Comment: P.7, l24: cululiform → cumuliform

**Response: (Page 9, Line 18)** We are sorry for our carelessness. In the revised paper, we have corrected the spelling problems.

31. Comment: P.7, l27: "has reached" → "is reached"

**Response: (Page 9, Line 22)** We are sorry for our carelessness. In the revised manuscript, we have corrected the grammar problems.

32. Comment: P.7, l30: exits → exists

**Response: (Page 9, Line 25)** We are sorry for our carelessness. In the revised manuscript, we have corrected the spelling problems.

33. Comment: P.8, l4: "when 1/2 for training." → "when 1/2 for training is used."

**Response: (Page 10, Line 3)** We are sorry for our carelessness. We have corrected this error in the revised manuscript.

34. Comment: P.8, l25: There is indeed improvement, but I would not call it "dramatically".

**Response: (Page 10, Line 23; Page 10, Line 29-33)** Many thanks for the constructive comment. We have deleted the word "dramatically" In the revised manuscript.

35. Comment: P.8, l26: What do you mean by "the statistical learning method"? I think this hasn't been defined before.

**Response: (Page 10, Line 24-25)** We are sorry not to make it clear. Statistical learning method is a framework for machine learning drawing from the fields of statistics and functional analysis (Trevor et al., 2009; Mohri et al., 2012). Statistical learning theory deals with the problem of finding a predictive function based on data. Because there is no definition before in the manuscript, we have changed this sentence as: "… on the other hand, the manifold features on the matrix manifold can describe the non-Euclidean geometric structure of the image features and thus …".

36. Comment: P.8, l32: Gabor or wavelet coefficients may need a citation, not generally known.

**Response: (Page 11, Line 1)** Many thanks for the constructive comment. We have added a reference Liu and Wechsler (2002) to make it clear in the revised manuscript.

37. Comment: P.9, l3: "on the both" → "on both,"

**Response: (Page 11, Line 6)** We are sorry for our carelessness. In the revised manuscript, we have corrected this problem.

38. Comment: Images and Tables: Please provide more telling captions. Table 4 and 5: The 1/10, 1/2 and so on are not clear.

**Response: (Page 17; Page 18)** Many thanks for the careful reading of our manuscript. We are sorry for our negligence. We have added some explanation of the fractions 1/10, 1 /2, 9/10 in Tables 4 and 5 in the revised manuscript to make it clear: 1/10, 1/2 and 9/10 are the certain proportions of the training set selected randomly from each category, and the rest part forms the testing set correspondingly.

***Special thanks for the reviewers' constructive comments.***

We have tried our best to improve the manuscript and made some changes in the manuscript. These changes will not influence the content and framework of the paper. And here we did not list the changes but marked in red or blue in the revised paper.

We appreciate for the reviewers' warm work earnestly, and hope that the correction will meet with approval.

Once again, thank you very much for your careful reading and inspiring suggestions.

**References**

Arsigny, V., Fillard, P., Pennec, X., and Ayache, N.: Geometric means in a novel vector space structure on symmetric positive-definite matrices, Siam J. Matrix Anal. A., 29, 328-347, doi:10.1137/050637996, 2008.

Calbó, J., and Sabburg, J.: Feature extraction from whole-sky ground-based images for cloud-type recognition, J. Atmos. Ocean. Techn., 25, 3, doi:10.1175/2007JTECHA959.1, 2008.

Cheng, H. Y., and Yu, C. C.: Block-based cloud classification with statistical features and distribution of local texture features, Atmos. Meas. Tech., 7, 1173-1182, doi:10.5194/amt-8-1173-2015, 2015.

Cristianini, N. and Shawe-Taylor, J.: An introduction to support vector machines and other kernel-based learning methods, Cambridge university press, 2000.

Gan, J., Lu, W., Li, Q., Zhang, Z., Yang, J., Ma, Y. and Yao, W.: Cloud type classification of total-sky images using duplex norm-bounded sparse coding, IEEE J. Sel. Top. Appl., 10, 3360-3372, doi:10.1109/JSTARS.2017.2669206, 2017.

Harandi, M. T., Sanderson, C., Wiliem, A., and Lovell, B. C.: Kernel analysis over Riemannian manifolds for visual recognition of actions, pedestrians and textures, In Proceedings of 2012 IEEE Workshop on the Applications of Computer Vision, Breckenridge, CO, USA, 433-439, 2012.

Heinle, A., Macke, A., and Srivastav, A.: Automatic cloud classification of whole sky images, Atmos. Meas. Tech., 3, 557-567, doi:10.5194/amt-3-557-2010, 2010.

Illingworth, A. J., Hogan, R. J., O'connor, E. J., et al.: Cloudnet: Continuous evaluation of cloud profiles in seven operational models using ground-based observations, B. Am. Meteorol. Soc., 88, 883-898, 2007.

Li, Q., Zhang, Z., Lu, W., Yang, J., Ma, Y. and Yao, W.: From pixels to patches: a cloud classification method based on a bag of micro-structures, Atmos. Meas. Tech., 9, 753-764, doi:10.5194/amt-9-753-2016, 2016.

Liu, C. and Wechsler, H.: Gabor feature based classification using the enhanced fisher linear discriminant model for face recognition, IEEE T. Image Process., 11, 467-476, doi:10.1109/tip.2002.999679, 2002.

Liu, S., Zhang, Z., and Mei, X.: Ground-based cloud classification using weighted local binary patterns, J. Appl. Remote Sens., 9, 095062, doi:10.1117/1.JRS.9.095062, 2015.

Mohri M., Rostamizadeh, A., and Talwalkar, A.: Foundations of machine learning, MIT press, Massachusetts, USA, 2012.

Ripley, B. D.: Pattern recognition and neural networks, Cambridge University Press, 8 edn., 2005.

Sanin, A., Sanderson, C., Harandi, M. T., and Lovell, B. C.: Spatio-temporal covariance descriptors for action and gesture recognition, In Proceedings of 2013 IEEE Workshop on Applications of Computer Vision (WACV), Tampa, FL, USA, 103-110, 2013.

Trevor, H., Robert, T., and Jerome, F.: The elements of statistical learning, Springer New York, 2009.

Tuzel, O., Porikli, F., and Meer, P.: Pedestrian detection via classification on Riemannian manifolds, IEEE T. Pattern Anal., 30,1713-1727, 2008.

Tzoumanikas, P., Kazantzidis, A., Bais, A. F., Fotopoulos, S., and Economou, G.: Cloud detection and classification with the use of whole-sky ground-based images, Atmos. Res., 113, 80-88, doi:10.1016/j.atmosres.2012.05.005, 2012.

Zhuo, W., Cao, Z., and Xiao, Y.: Cloud classification of ground-based images using texture-structure features, J. Atmos. Ocean. Techn., 31, 79-92, doi:10.1175/JTECH-D-13-00048.1, 2014.

---

## Author Response (AR2)

**Response to editor's comments**

We thank the editor for the insightful and thoughtful comments, which have allowed us to produce a stronger manuscript. Our responses to the comments are given below, and the corresponding changes are tracked at specific location (Page X, Line X) in the revised manuscript.

**Comments to the Author:**

I have read your response to the referees and the corrections to the revised manuscript and find it is acceptable for publication pending further attention to some minor details:

1. p2: owns several advantages -> has several advantages

**Response: (Page 2, Line 22)** Many thanks for the careful reading of the manuscript. We have corrected it.

2. p9 line 26: especially when -> even when only

**Response: (Page 9, Line 7)** Many thanks for the careful reading of the manuscript. We have corrected it.

3. p9 line 28: suggested wording... add [xxxxxxx] ... delete /xxxxxx/
   ... gained even with [what would normally be regarded as] insufficient ... /as well/

**Response: (Page 9, Line 9-10)** Many thanks for the constructive comment of the manuscript. We have corrected it as suggested.

4. p9 and p10: suggest avoiding the use of vague terms such as ... insufficient, enough, quite enough

**Response: (Page 9, Line 11; Page 9, Line 32-Page 10, Line 3)** Many thanks for the constructive comment of the manuscript. Here we use "insufficient, enough, quite enough" is to explain that with different fractions of data reserved for training, the performance varies. To make it clear, we use these vague terms as little as possible.

5. p9 line 29: when -> where
   ... where the fractions of data reserved for training constitute 1/10, 1/2, and 9/10 of the total...and are chosen to represent a wide range of possible training scenarios.

**Response**: **(Page 9, Line 30-32)** Many thanks for the careful reading of the manuscript. We have corrected it.

6. p9 line 32: not sure if "typical" refers to the 1/10 case and "sufficient enough" is 1/2. please clarify.

**Response: (Page 9, Line 13-14)** Many thanks for the constructive comment of the manuscript. Here

"typical" means that the images are representative of a wide range of possible training scenarios to ensure the performance of the algorithm. "sufficient enough" is to mean that more images are used for training to make the training set adequate to improve the performance of the algorithm. To make it clear, this sentence has changed as "As a result, as more representative training images are used for training, there is no doubt that the recognition rate will be improved."

7. p10 line 9: suggest adding ...
   ... on the whole when compared to expert meteorological analysis

**Response: (Page 9, Line 23)** Many thanks for the insightful comment of the manuscript. We have added this part in the manuscript.

8. Tables 4 and 5 can delete "part".

**Response: (Page 18-19)** Many thanks for the careful reading of the manuscript. We have deleted this word.

***Special thanks for the reviewers' and Editor's constructive comments.***

We have tried our best to improve the manuscript according to the insightful comments. The changes made in the manuscript will not influence the content and framework of the paper. And here we did not list the changes but marked in the revised paper.

We appreciate for the reviewers' and editor's warm work earnestly, and hope that the correction will meet with approval.

Once again, thank you very much for your careful reading and inspiring suggestions.

[revised manuscript text omitted]